# Comparison of Phenolic Compounds and Evaluation of Antioxidant Properties of *Porophyllum ruderale* (Jacq.) Cass (*Asteraceae*) from Different Geographical Areas of Queretaro (Mexico)

**DOI:** 10.3390/plants12203569

**Published:** 2023-10-14

**Authors:** Ángel Félix Vargas-Madriz, Ivan Luzardo-Ocampo, Jorge Luis Chávez-Servín, Ulisses Moreno-Celis, Octavio Roldán-Padrón, Haidel Vargas-Madriz, Haydé Azeneth Vergara-Castañeda, Aarón Kuri-García

**Affiliations:** 1Department of Cell and Molecular Biology, School of Natural Sciences, Universidad Autonoma de Queretaro, Qro 76230, Mexico; angel.vargas@uaq.mx (Á.F.V.-M.); jorge.chavez@uaq.mx (J.L.C.-S.); ulisses.moreno@uaq.mx (U.M.-C.); octavio.roldan@uaq.mx (O.R.-P.); 2Tecnologico de Monterrey, The Institute for Obesity Research, Ave. Eugenio Garza Sada 2501 Sur, Monterrey 64841, Mexico; ivan.8907@gmail.com; 3Tecnologico de Monterrey, School of Engineering and Sciences, Campus Guadalajara, Av. General Ramon Corona 2514, Zapopan 45201, Mexico; 4Department of Agricultural Production, Centro Universitario de la Costa Sur, University of Guadalajara, Av. Independencia Nacional, No. 151, Autlán 48900, Mexico; haidel_vargas@hotmail.com; 5Advanced Biomedical Research Center, School of Medicine, Universidad Autonoma de Queretaro, Qro 76140, Mexico; hayde.vergara@uaq.mx

**Keywords:** *Porophyllum ruderale*, antioxidant capacity, bioactive compounds, phenolic compounds

## Abstract

*Porophyllum ruderale* (*P. ruderale*) is a well-known Mexican plant from the group of “Quelites”, widely consumed plant species used for several food and medicinal purposes. As the production is very heterogeneous and the diverse agroclimatic conditions significantly impact the plant’s phytochemical composition, this research aimed to compare the phenolic compound composition and the antioxidant capacity of the *P. ruderale* plant from three different collection sites (Queretaro, Landa de Matamoros, and Arroyo Seco) in the State of Queretaro (Mexico). Plants collected from Queretaro displayed the lowest total phenolic compounds, flavonoids, and condensed tannins, reflected in a lower antioxidant capacity (DPPH, FRAP, ABTS), compared to the other collection places. Flavones (epicatechin and epigallocatechin gallate) were the most abundant (36.1–195.2 μg equivalents/g) phenolics quantified by HPLC-DAD, while 31 compounds were identified by UHPLC-DAD-QToF/MS-ESI. Most compounds were linked to biological mechanisms related to the antioxidant properties of the leaves. A PCA analysis clustered Landa de Matamoros and Arroyo Seco into two groups based on flavones, hydroxybenzoic acids, the antioxidant capacity (ABTS and DPPH), and total phenolic compounds, the main contributors to its variation. The results indicated contrasting differences in the polyphenolic composition of collected *P. ruderale* in Queretaro, suggesting the need to standardize and select plants with favorable agroclimatic conditions to obtain desirable polyphenolic compositions while displaying potential health benefits.

## 1. Introduction

*Porophyllum ruderale* (Jacq.) Cass. (Asteraceae) (*P. ruderale*) is an annual herbaceous plant species native to the Western Hemisphere whose taxon occurs in North, Central, and South America. In different parts of Mexico and South America, it is known as “papalo-quelite”, “tepelcasho”, and “tepegua” [1]. *P. ruderale* belongs to the “Quelites” group of plants, a vast term derived from the ancient Nahuatl word “Quilitl”, meaning edible grass, and since pre-Hispanic times, these plants have been a fundamental part of the traditional Mexican diet. In Mexico, about 500 different species of quelites are consumed, but only half of them have been properly studied [2]. In rural areas, quelites form part of a balanced diet because of their practical harvest and easy preparation (raw or cooked) in several dishes for human consumption, contributing to essential nutrients intake [3]. However, changes in social dynamics, environmental conditions, and dietary inclusion of food products from the so-called “Western diet” provoked the disappearance of some of these plants or subjected them to marginal production, commercialization, and consumption [4]. 

*P. ruderale* is one of the most consumed quelites, with a flavor described as between arugula (*Eruca vesicaria*), cilantro (*Coriandrum sativum*), and rue (*Ruda graveolens*) [5]. The leaves and stems of *P. ruderale* are used in traditional cooking as a spice for raw and cooked foods such as salads or stews. In addition, in traditional medicine, *P. ruderale* is believed to relieve general pain and participate in closing wounds, with biological properties that might be attributed to bioactive compounds, mainly polyphenols [1]. Phenolic compounds are well-known plant secondary metabolites with a wide range of health-associated properties, mainly used as antioxidants in both the pharmaceutical and food industries due to their inhibitory effect on the propagation of free radicals [6]. In Mexico, *P. ruderale* harvest reached more than 8000 tons by 2021, and the primary producing States were Guerrero (4597 tons), Morelos (1210), and Puebla (1008 tons) [7]. Other states, such as Queretaro, share the remaining production, where *P. ruderale* is considered part of the native flora [8].

Few studies have assessed the nutritional composition and the phytochemical composition of *P. ruderale* plants, particularly considering factors such as the geographical origin at which the plants are harvested and collected since the soil composition, harvest time, plant maturity, agroclimatic conditions, and extraction methods can alter their phytochemical composition [9]. The broad ability of *P. ruderale* plants to adapt to several environmental conditions can modify the concentration of phenolic compounds, involving complex mechanisms of physiological action and molecular programs related to several genes and pathways [10]. Considering that *P. ruderale* leaves are used in traditional cooking as a food additive and that there is an increase in studies related to the plant’s cultivation, nutrition, and pharmacology, it would be useful to determine the variability of *P. ruderale* polyphenolic composition in several geographical locations of Queretaro (Mexico). Therefore, this research aimed to compare the phenolic compounds and antioxidant capacity of *P. ruderale* leaves collected in three geographical areas of Queretaro. 

## 2. Results

### 2.1. Identification and Characterization of the Polyphenolic Content of Porophyllum ruderale Samples and Antioxidant Capacity

Once representative places were selected for the *P. ruderale* leaves collection, the leaves were further processed to spectrophotometrically quantify total free phenolic compounds and the antioxidant capacity of the prepared extracts. Although Arroyo Seco samples displayed the highest total phenolic compounds (TPC) and condensed tannins (CT) amount, Landa de Matamoros samples displayed the highest (*p* < 0.05) amount of total flavonoids (TF) (Table 1). Samples from Queretaro exhibited the lowest TPC contents, similar values of TF to Arroyo Seco, and similar CT to Arroyo Seco. Surprisingly, Landa de Matamoros showed the highest 2,2-diphenyl-1-picrylhydrazil (DPPH), ferric reducing antioxidant power (FRAP), and 2,2-azino-bis(3-ethylbenzothiazoline-6-sulfonic acid) (ABTS) antioxidant capacity. 

Table 2 shows the ultra-high-performance chromatographic analysis, coupled with diode-array-detector, quadrupole time of-flight, and mass spectrometry with electrospray ionization (UHPLC-DAD-QToF/MS-ESI) characterization of the plant extracts from the three locations. As observed, seven hydroxycinnamic acids and derivatives, nine flavonols, three flavones, two flavanones, and four compounds that were identified as “other compounds” were found.

Along with UHPLC characterization, selected phenolic compounds were quantified by high-performance liquid chromatography (HPLC) analysis, and an enrichment pathways metabolomic analysis was conducted to calculate the statistical significance of differences between metabolites. Since Landa de Matamoros and Arroyo Seco displayed the highest amount of compounds, these samples were considered for the HPLC characterization (Figure 1A). For flavonols, particularly epicatechin and (+)-catechin, Landa de Matamoros samples had 23.30% higher epicatechin content than Arroyo Seco samples, and Arroyo Seco exhibited 44.96% more (+)-catechin than Landa de Matamoros samples (Figure 1A, Appendix A). In contrast, the samples contained hydroxycinnamic acids and derivatives, hydroxybenzoic acids and derivatives, and benzaldehydes and benzenoids values, which ranged between 0.6 and 12.3 μg/g (Appendix A). The enrichment pathways metabolomic analysis (Figure 1B) highlighted that, collectively, the identified and quantified compounds mainly belonged to hydroxybenzoic and m-methoxybenzoic acids pathways, whereas flavonoids-associated pathways (flavones, flavonoid glycosides; flavans, flavanols, and leuchoanthocyanins; and flavanones) displayed the lowest *p*-values. 

### 2.2. Bioinformatic Analysis of Health-Associated Pathways Linked to Polyphenolic Composition

Bioinformatic analysis was conducted to elucidate potential health-associated pathways associated with the polyphenolic composition of the samples (Figure 2). The main pathways in which the *P. ruderale* could be causing an impact were related to several mechanisms, mainly those related to lipids metabolism, amino acids metabolism, xenobiotic biodegradation and metabolism, replication and repair, signal transduction, cell growth and death, immune and endocrine systems, nervous system, aging, several cancer types, cardiovascular diseases, and antineoplastic drug resistance. 

### 2.3. Principal Components Analysis (PCA) and Correlations between the Polyphenolic Composition and the Antioxidant Capacity of the Samples

A PCA of the samples is shown in Figure 3, where two of the principal components (PC) explained 97.78% of the total variation in the samples (Appendix A). In these components, such as (+)-catechin, epicatechin, epigallocatechin gallate, and hydroxybenzoic acids and their derivatives, e.g., gallic, hydroxybenzoic, and hydroxyphenylacetic acids, the antioxidant capacity of ABTS and DPPH, TPC, and TF explained the total variation in PC1. In contrast, CT, TPC, and FRAP antioxidant capacity and a flavonol (rutin) mostly explained the total variation in PC2 (Figure 3A). The samples were grouped in two differentiated clusters (Figure 3B), with TPC, rutin, epigallocatechin, and gallic acid having the greatest effect in samples from Arroyo Seco, while epicatechin, ferulic acid, and DPPH antioxidant capacity showed the same effect in samples from Landa de Matamoros (Figure 3B). In addition, as shown in the loading plots (Figure 3B), the proximity of TPC to epigallocatechin gallate, gallic acid, and rutin suggests their relatedness. In contrast, ferulic acid and epicatechin are suggested to be the main contributors to DPPH antioxidant capacity. Although three components were indicated in Figure 3A, only two of them were plotted in Figure 3B as just two of them were enough to explain >80% of the total variation. Moreover, dispersion of the samples along PC2 (Figure 3B) could be associated with its contrasting TPC and selected HPLC-DAD quantified contents.

Spearman correlations between the quantified polyphenols by HPLC and spectrophotometrically and the antioxidant capacity are shown in Figure 3. Similar to the results presented in Figure 1, some hydroxycinnamic acids and derivatives such as ferulic acid, *p*-coumaric acid, and flavonols such as quercetin and epicatechin were the highest correlated with the antioxidant capacity by ABTS, DPPH, and FRAP. In addition, TPC was positively correlated with sinapic acid, gallic acid, rutin, and epigallocatechin gallate. 

## 3. Discussion

This research was intended to compare the composition of phenolic compounds and the antioxidant capacity of *P. ruderale* leaves collected in three different locations from the State of Queretaro (Mexico), one of the states involved in the marginal production of this quelite. The regions were selected based on contrasting climate conditions, which are known to stimulate the production of secondary metabolites in plants, such as phenolic compounds, most of which are accumulated in the plants’ vacuoles, protecting the plant against not only abiotic but also biotic stresses [11]. Hence, *P. ruderale* leaves from three representative climate zones of the State of Queretaro were collected, such as Tlacote El Bajo (municipality of Queretaro), a dry, warm, and arid scrub zone (xerophilic flora), and a semiwarm and semihumid temperature zone in El Aguacate (municipality of Landa de Matamoros) and Acatitlán de Zaragoza (municipality of Arroyo Seco) [12]. The particularities of each agroclimatic region could impact the contrasting differences between TPC, TF, and CT of the plants, as high-temperature conditions are known to increase phenolic compounds in plant leaf tissue [13], which agreed with our results, as Landa de Matamoros (12–24 °C) has a relatively higher average temperature than Arroyo Seco (18–22 °C). However, these results must be taken with caution as temperature is not the only factor affecting the composition and synthesis of secondary metabolites in plants, and average temperatures change during seasons [14]. Other factors known to affect the concentration of secondary metabolites in plants are water availability, UV exposure, soil nutrients, and moisture, among others [10]. Compared to arid zones, temperate forest and jungle areas have a wider water availability, providing a great amount of oxygen and better soil nutrient composition with a variety of microorganisms known to biotransform isoflavones and flavanols into nitrogen to be fixed in plants as nodules, which also includes the production of flavonoids and phenolic acids [10,15]. 

Lower TPC values (−26.38 to −52.92%) were reported by Conde-Hernández and Guerrero-Beltrán [1] in *P. ruderale* plants from the center-south of Mexico (Atlixco, Puebla) after an acidified-ethanol extraction (85:15 ethanol:HCl) and ultrasonication/stirring, despite the acidic conditions being a method causing hydrolysis of components in the plant matrix, which can be explained by the higher concentration achieved in our research after freeze-drying. More recently, Fukalova et al. [4] also reported lower TPC contents (−91.9%), but these results were based on fresh weight, a less accurate method for reporting phenolics, as water highly varies on plant leaves. Overall, *P. ruderale* leaves contain a greater number of phenolic compounds and higher antioxidant capacity than other quelites such as “quelite cenizo” (*Chenopodium berlandieri* spp. *Berlandieri*) and “quintonil” (*Amaranthus hybridus*), another well-known quelite [16]. 

In this study, additional phenolic compounds such as isoferulic/ferulic acids, sinapic acid, protocatechuic acid (3,4-dihydroxybenzoic acid), epigallocatechin, myricetin, apigenin, pinocembrin, and naringenin were found for *P. ruderale* compared with comprehensive studies that reported the complete polyphenolic and metabolomic characterization of *P. ruderale* leaves [17,18], providing more information about the highly variated polyphenolic composition of *P. ruderale* cultivated worldwide. However, most authors agree that *P. ruderale* leaves primarily contain hydroxycinnamic acids (caffeic and chlorogenic acids derivatives) and quercetin and kaempferol complexes with galactosides. A high abundance of hydroxycinnamic and hydroxybenzoic acids was highlighted in the enrichment metabolomic pathways analysis, which is a metabolomics analysis approach using the Mummichog algorithm, allowing us to predict the functional activity of metabolites through their correspondence to the global network of plant metabolic pathways [19]. However, although flavonoids are the most abundant compounds, as indicated in Table 1, other phenolic classes are more diverse in *P. ruderale* leaves (Table 2, Figure 1). 

Overall, the total polyphenolic composition contributed to its antioxidant capacity, which is mainly responsible for most of the biological pathways that could be modulated by *P. ruderale* phenolics. The involvement of these phenolics in the highlighted conditions and pathways in Figure 2 are based on a prediction between the tested phenolics and the main human biological and signaling pathways using neural networks [20]. Particularly, in lipid metabolism, phenolic compounds participate in antioxidative mechanisms reducing the activation of the arachidonic acid cascade by inhibiting or reducing cyclooxygenase (COX) and lipoxygenase (LOX) activation, thus showing beneficial effects in reducing the risk of cardiovascular disease and cancer [21]. Regarding amino acid metabolism, polyphenols can inhibit tryptophan breakdown in peripheral blood mononuclear cells and regulate its depletion, thus potentially preventing neural diseases such as Alzheimer’s and Huntington’s disease [22]. 

Despite the predictable association between *P. ruderale* polyphenols and biological mechanisms, the results must be taken cautiously, as in vivo and in vitro validation must be carried out. Although there is no information about the scientific examination of *P. ruderale* biological activity, our research group previously showed that *P. ruderale* leaves can reduce the metabolic activity of human SW480 colorectal cancer cells through a cytotoxic effect attributable to its polyphenolic composition, particularly rutin, epigallocatechin gallate, caffeic acid, and gallic acid [23]. Other authors reported the anti-inflammatory potential of *P. ruderale* extracts and attributed most of the effects to chlorogenic acid, several flavonoids, and ellagitannins [18]. In addition, the way the plants are consumed or their compounds extracted is critical since dried *P. ruderale* plants showed low antioxidant capacity and a low correlation between TPC and ABTS [1]. However, identifying composition differences between plants might serve to potentially modulate their metabolites’ composition, considering the local conditions under which the plants are cultivated. In this research, *P. ruderale* leaves from Arroyo Seco showed a distinctive and separated clustering from the Landa de Matamoros collection site (Figure 3), and, predictively, most of the polyphenolic composition could correlate with the antioxidant capacity methods, which has also been exhibited in leaves such as *Moringa oleifera,* where the measurement of several hydroxycinnamic classes highly correlated with ABTS and DPPH. These results are even more similar to an in vivo approach as the measurement was considered after in vitro gastrointestinal digestion of the leaves [24]. 

## 4. Materials and Methods

### 4.1. Geographical Location of Porophyllum ruderale Sampling Sites

The State of Queretaro is characterized by a dry and semidry climate located in the central region (about 51% of the state’s surface); a warm, semihumid climate in the Sierra Madre Oriental region (24.3% of the state’s surface); a temperate, subhumid climate in the southern, central, and northeastern regions (23% of the state’s surface); a humid, warm climate towards the northeast (1%); and a humid, temperate climate in the northeast region (0.7%). Moreover, an arid scrub zone predominates in the central part of the state, followed in importance by coniferous and holm oak forests, in the high areas of the north, and deciduous jungle [12]. Within the state, three municipalities were selected as sampling sites (Queretaro, Arroyo Seco, and Landa de Matamoros), because each is representative of the state’s climate. Queretaro is characterized by primarily dry and temperate weather, and Arroyo Seco also contains characteristics of a semiwarm climate, whereas in Landa de Matamoros, a temperate humid, and subhumid environment is shown (Figure 4).

### 4.2. Plant Material Collection and Processing

*Porophyllum ruderale* (Jacq.) Cass. (Asteraceae) (The International Compositae Alliance (TICA) checklist record: D78AD427-6E4D-4C8C-8C6A-BF2EDC6EA06C) fresh leaves were collected from three localities in the State of Querétaro (Mexico): (1) Tlacote El Bajo (municipality of Querétaro) (GPS coordinates: longitude (dec): −100.507222, latitude (dec): 20.662222) at 1850 m.a.s.l. (2) Acatitlán de Zaragoza (municipality of Landa de Matamoros) (GPS coordinates: longitude (dec): −99.188056, latitude (dec): 21.206389) at 1200 m above sea level (m.a.s.l.), with cloud forest vegetation. It had an average annual precipitation of 491.70 mm (2022), has an altitude of 1288 m.a.s.l., and has a semiwarm, subhumid climate [25]. (3) El Aguacate (municipality of Arroyo Seco) (GPS coordinates: longitude (dec): −99.616667, latitude (dec): 21.399722) at 740 m.a.s.l. Its vegetation is low deciduous forest and xeric vegetation, with an annual precipitation of 274 mm (2022) and a semidry climate [26,27]. For all cases, a total of 15 samples were collected in areas associated with seasonal agricultural production in the dry season (spring/summer) of the Sierra Gorda and Querétaro regions, for which the samples collected were associated with a humid climate and a dry climate, respectively. The leaves were identified by a specialist from the Jerzy Rzedowski Herbarium of the Universidad Autónoma de Queretaro and sampled using the recommended acronyms of Thiers [28] as QMEX157. Once they arrived at the laboratory (all samples were processed 4 h after collection to standardize the transportation time from the collection sites to the laboratory), the leaves were processed as previously indicated in Vargas-Madriz et al. [23]. Briefly, a forced-air-drying process at 40 °C for 72 h was conducted in an oven (FX 1375, Shel-Lab, Elkin, NC, USA), followed by grinding in an electric mill, sieving (0.5 mm particle size), storage in airtight bags, and freezing (Last II, REVCO, Twinsburg, OH, USA) at −80 °C.

### 4.3. Preparation of Plant Extracts, Identification, and Quantification of Free Phenolic Compounds

The extracts were prepared using the processed samples of *P. ruderale*, mixed with a 80:20 (*v*/*v*) ethanol:water solution in a powder:water ratio of 1:10, and shaken (100 rpm, 22 ± 1 °C) for 16 h protected from light, filtered (Whatman paper, 0.22 μm) (Sigma-Aldrich, St. Louis, MO, USA), freeze-dried for 72 h in a freeze-dryer (FreeZone 6.0 Benchtop Free Dry System, Labconco, Kansas City, MO, USA), and stored at −80 °C. The lyophilized powders were diluted in 100% methanol at 1 mg dry lyophilized extract/mL for each spectrophotometric or chromatographic determination. 

The spectrophotometric quantification of free phenolic compounds was performed using the Folin–Ciocalteu procedure for total phenolic compounds, and the results were indicated in milligrams of gallic acid equivalents (GAE)/100 g lyophilized extract [29], using a gallic acid (Sigma-Aldrich) standard curve. The total flavonoid content was determined using NaNO_2_, AlCl_3_, and NaOH as the main reagents [30] and using a (+)-catechin (Sigma-Aldrich) standard curve. Total condensed tannins were also quantified through a (+)-catechin standard curve using a colorimetric vanillin method [31].

The chromatographic quantification of selected phenolic compounds was conducted using a high-performance liquid chromatography (HPLC) system, as reported in Sánchez-Quezada et al. [32]. Briefly, an Agilent 1100 Series HPLC System (Agilent Technologies, Palo Alto, CA, USA) coupled with a diode-array detector (DAD) was used. A Zorbax Eclipse XDB-C18 column (Agilent Technologies) (4.6 × 250 mm, 5 μm granule size) was used at a flow rate of 0.750 mL/min and 35 ± 0.5 °C temperature (maximum pressure: 400 bar). A 20 μL sample volume was injected, previously filtered through 0.45 μm acrodiscs filters. The mobile phase consisted of solvent A: acidified water with formic acid at 0.1%, and solvent B: 100% acetonitrile, both HPLC-grade reagents (Prolab company, Qro, Mexico). A linear gradient was used at the indicated time steps: 100% A (0 min), 95% A (3 min), 88% A (4 min), 80% A (7 min), 78% A (13 min), 77% min (14 min), 75% A (15 min), 74% A (17 min), 72% A (20 min), 70% A (22 min), 50% A (25 min), 25% A (26 min), and 0% A (27 min). The detection was conducted at 240, 280, and 360 nm. Standard curves of HPLC-grade reagents such as chlorogenic acid, sinapic acid, caffeic acid, *p*-coumaric acid, ferulic acid, gallic acid, hydroxybenzoic acid, hydroxyphenylacetic acid, rutin, quercetin, (+)-catechin, epicatechin, and epigallocatechin gallate (Sigma-Aldrich) were used. The results were expressed in micrograms equivalents of each compound/g sample, but for comparison purposes, the samples were normalized using a min-max normalization as follows: normalized content = (amount of selected compound-minimum content in the samples)/(maximum content − minimum content).

In addition to spectrophotometry and HPLC-DAD analysis, an ultra-high-performance chromatographic analysis, coupled with diode-array-detector, quadrupole time-of-flight, and mass spectrometry with electrospray ionization (UHPLC-DAD-QToF/MS-ESI) was followed as reported by Sánchez-Recillas et al. [33]. A BEH Acquity C18 column (2.1 × 100 mm, 1.7 μm granule size) was used at 35 ± 0.5 °C. Samples were separated using the same solvents as in the HPLC-DAD analysis, but different gradient conditions were followed (0.4 mL/min, injection volume: 2 μL): 95% A (0 min), 95% A (2 min), 5% A (22 min), 5% A (25 min), 5% A (27 min), and 5% A (30 min). Selected absorbance channels (214, 280, 320, and 360 nm) were used (Appendix A). 

The ESI ionization was conducted in positive and negative scan modes. For the MS analysis, parameters were set as follows: low-collision energy: 6 eV, high-collision energy: 15–45 eV, mass range: 50–1800 *m*/*z*, capillary voltage: 2.0 and 3.5 kV for the negative and positive scans, respectively, source temperature: 120 °C, desolvation temperature: 450 °C. Argon was used as the collision and desolvation gas at 50 L/h and 800 L/h, respectively, using 40 V as cone voltage. Mass correction was achieved through a leucine-enkephalin solution (50 pg/mL) at 10 mL/min. The exact mass of the pseudo-molecular ions (mass error < 10 ppm), isotope distribution, and fragmentation patterns were used to identify each compound. Data were analyzed with UNIFI v. 1.9, Service Release 4 software (Waters Co., Milford, MA, USA). The fragments were identified using the reported fragmentation patterns from PubChem, FooDB v. 1.0, HMDB v. 5.0, and MassBank of North America (MoNA). 

### 4.4. Antioxidant Capacity Quantification

The antioxidant capacity was measured as the ability of the plant extracts to scavenge several test radicals. Hence, the 2,2-diphenyl-1-picrylhydrazil (DPPH) assay was performed as reported by Fukumoto and Mazza [34]. The ferric reducing antioxidant power (FRAP) assay was conducted following the procedure of Benzie and Strain [35]. The 2,2-azino-bis(3-ethylbenzothiazoline-6-sulfonic acid) (ABTS) assay was conducted as reported by van den Berg et al. [36]. Trolox was used as control for all the experiments, and a Trolox standard curve was prepared to express the antioxidant capacity as mg equivalents of Trolox/g sample. 

### 4.5. In Silico Analysis of Metabolic Pathways and Potential Beneficial Effects

An in silico analysis of the molecular pathways prediction was conducted using the PathwayMap utility from PlayMolecule (https://playmolecule.org, accessed on 26 August 2023). Moreover, data were also analyzed using the Enrichment Pathways Analysis feature of MetaboAnalyst 5.0 (https://www.metaboanalyst.ca/, accessed on 26 August 2023). 

### 4.6. Statistical Analysis

Data were presented as the mean ± SD of three independent experiments in triplicates. An analysis of variance (ANOVA) was conducted, followed by a post hoc Tukey-Kramer’s test, were *p* < 0.05 was considered significant. Statistical analysis was performed in JMP v. 17.1.0 (SAS, Cary, NC, USA).

## 5. Conclusions

The results obtained from this research suggested that *P. ruderale* plants display a contrasting polyphenolic composition, which may influence their antioxidant capacity and their potential biological activity in molecular mechanisms related to this property. Since the clustering plotting after principal components analysis distinguished two different collection sites (Landa de Matamoros and Arroyo Seco), it might be inferred that agroclimatic conditions are desired for phytochemicals’ accumulation, indicating future needs in the standardization of phenolic compounds’ contents. Moreover, this research provides evidence about the chemical composition of native plants for further inclusion in marketable products with added value. 

## Figures and Tables

**Figure 1 plants-12-03569-f001:**
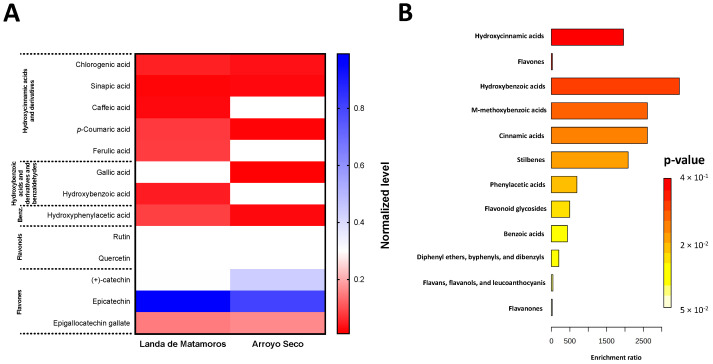
Normalized HPLC composition (**A**) and enrichment metabolic analysis (**B**) of identified and quantified phenolic compounds in *P. ruderale* samples from Landa de Matamoros and Arroyo Seco collecting sites. Benz: benzenoids. HPLC composition was normalized using a min-max normalization. For the exact HPLC-DAD values, please refer to Appendix A.

**Figure 2 plants-12-03569-f002:**
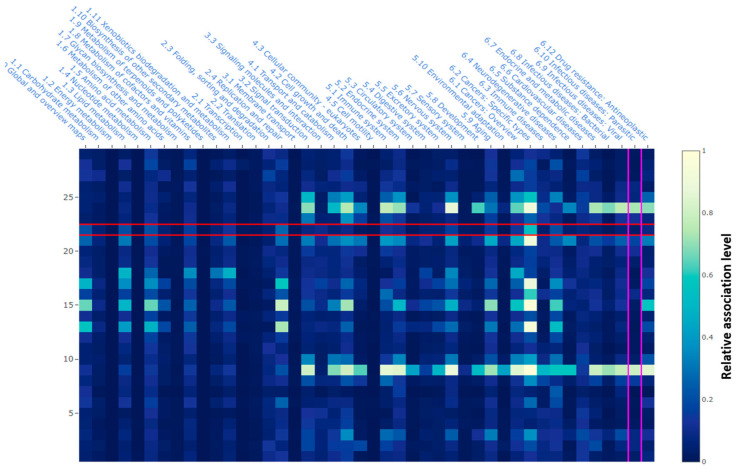
Bioinformatics analysis of pathways linked to the polyphenolic composition of *P. ruderale* samples. The figure was generated in the PathwayMap module from https://playmolecule.org (accessed on 21 July 2023). The red and purple lines are an sample indication of matching between a predicted metabolic pathway and its relative association.

**Figure 3 plants-12-03569-f003:**
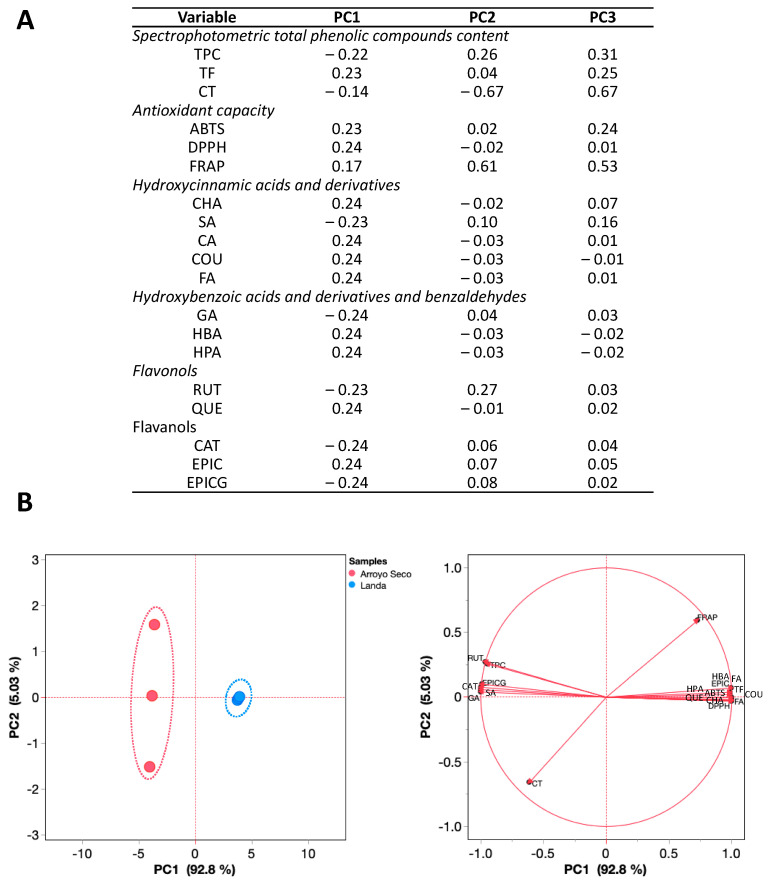
Principal components analysis (PCA) of the polyphenolic composition (HPLD-DAD) and the antioxidant capacity of *P. ruderale* samples from Arroyo Seco and Landa de Matamoros. (**A**) Participation of each principal component (PC) in the total variation. (**B**) Scatter and loading plots of the first and the second principal components for all the assessed samples. ABTS: 2,2-azino-bis(e-ethylbenzotiazoline-6-sulfonic acid); CA: caffeic acid; CAT: (+)-catechin; CHA: chlorogenic acid; COU: *p*-coumaric acid; CT: condensed tannins; DPPH: 2,2-diphenyl-1-picrylhydrazil; EPIC: epicatechin; EPICG: epigallocatechin gallate; FA: ferulic acid; FRAP: ferric reducing antioxidant power; GA: gallic acid; HBA: hydroxybenzoic acid; HPA: hydroxyphenylacetic acid; RUT: rutin; QUE: quercetin; SA: sinapic acid; TF: total flavonoids; TPC: total phenolic compounds.

**Figure 4 plants-12-03569-f004:**
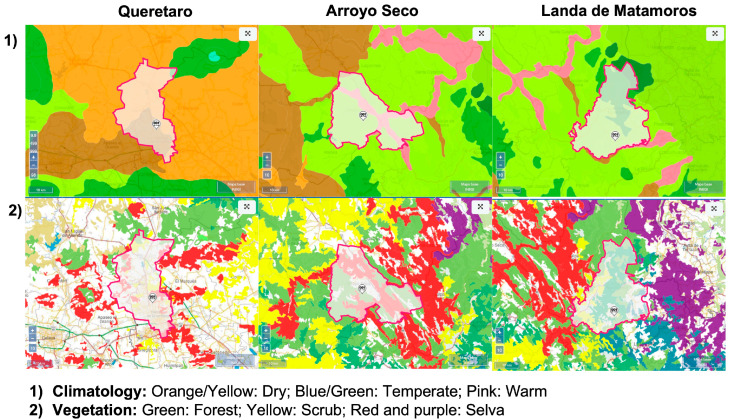
(**1**) Climatology and (**2**) use of soil land and vegetation of the sampled areas from the state of Queretaro (Mexico). A 10 km scale was used to visualize pictures taken from the Mexican National Institute of Statistics and Geography (Spanish acronym: INEGI) [12].

**Table 1 plants-12-03569-t001:** Spectrophotometric total phenolic compounds content and antioxidant capacity of *Porophyllum ruderale* samples from three different geographical areas of Queretaro.

Parameter	Queretaro	Landa de Matamoros	Arroyo Seco
Spectrophotometric total phenolic compounds content			
TPC (mg GAE/100 g)	5555.00 ± 242.10 ^c^	7382.00 ± 233.70 ^b^	8687.00 ± 259.50 ^a^
TF (mg CE/100 g)	4579.00 ± 275.50 ^b^	9291.00 ± 702.60 ^a^	4748.00 ± 262.50 ^b^
CT (mg CE/100 g)	2.70 ± 0.10 ^b^	2.90 ± 0.06 ^b^	3.00 ± 0.10 ^a^
Antioxidant capacity (mg TE/100 g)			
DPPH	254.00 ± 3.40 ^b^	561.00 ± 4.30 ^a^	256.00 ± 2.70 ^b^
FRAP	511.00 ± 35.50 ^c^	698.00 ± 28.60 ^a^	635.00 ± 47.10 ^b^
ABTS	422.00 ± 35.60 ^c^	573.00 ± 13.30 ^a^	485.40 ± 3.70 ^b^

The results are expressed as the mean ± SD of three independent experiments in triplicate. Different letters by row express significant differences (*p* ≤ 0.05) by Tukey–Kramer’s test. ABTS: 2,2-azino-bis(3-ethylbenzothiazoline-6-sulfonic acid); CE: (+)-catechin equivalents; CT: condensed tannins; DPPH: 1,1-diphenyl-2-picrylhydrazil; FRAP: ferric reducing antioxidant power; GAE: gallic acid equivalents; TE: Trolox equivalents; TF: total flavonoids; TPC: total phenolic compounds.

**Table 2 plants-12-03569-t002:** UHPLC-DAD-QToF/MS-ESI characterization of identified phenolic compounds and other components of *P. ruderale* leaves from Landa de Matamoros and Arroyo Seco collection sites.

Ionization Mode	Compound Name	RT (min)	Expected Mass (Da)	Observed *m*/*z*	Mass Error (ppm)	Adducts	Fragments
Hydroxycinnamic acids and derivatives							
ESI–	Trans-cinnamic acid	6.99	148.0524	148.0523	–0.5	[M-H]^−^	147.04447, 103.0553, 101.02457
ESI–	*p*-Coumaric acid	7.91	167.0574	164.0473	0.08	[M-H]^+^	119.05035, 145.03, 117.03462
ESI-	Isoferulic acid	8.16	194.0579	194.058	0.3	[M-H]^−^	135.04484, 134.03711, 133.02948, 132.02168
ESI–	Ferulic/trans ferulic acid	8.5	194.0579	194.058	–0.6	[M-H]^−^	149.02377, 175.00367, 121.02954, 193.01372
ESI–	Sinapic acid	9.26	224.0685	224.0685	–0.3	[M-H]^−^	193.01372, 164.01078, 149.0238, 163.03919
ESI–	Protocatechuic acid	7.8	154.0262	154.0267	–2.7	[M-H]^−^	109.02928, 108.0216, 81.03493
ESI–	Chlorogenic acid	7.76	354.095	354.0954	–0.2	[M-H]^−^	191.05603, 161.02435, 135.04497, 93.03474, 85.02967
Flavonols							
ESI–	Quercetin-3-*O*-rhamnosyl-galactoside	8.55	610.1533	610.1525	–2.1	[M-H]^−^	301.03408, 609.14196, 300.02701, 255.0301
ESI+	Fisetin	10.18	286.0475	286.0474	–0.8	[M+H]^+^	285.03998, 286.0435, 163.00372, 135.008
ESI–	Rutin	9.08	610.1534	610.1545	–0.7	[M+H]^−^	271.0968, 609.14334
ESI–	Quercetin	11.01	302.0421	302.0428	0.4	[M-H]^−^	301.03454, 151.00371, 178.99852, 121.02931, 107.01382
ESI–	Epigallocatechin	6.99	306.0725	306.0725	–4.8	[M-H]^−^	165.07467, 125.02448, 109.02842, 139.08699
ESI–	Myricetin	7.45	318.0379	318.0379	1	[M-H]^−^	137.02512, 109.02808, 178.02227
ESI+	Kaempferol-3-*O*-rutinoside (Nicotiflorin)	9.96	594.1571	594.1586	–2.3	[M+H]^+^	255.02954, 285.03945, 284.03225, 593.43848, 227.03464
ESI–	Kaempferol-3-*O*-β-D-glucoside (Astragalin)	9.64	448.1005	448.1009	0.1	[M-H]^−^	255.02962, 447.09349, 284.03251, 227.03474
ESI+	Kaempferol	11.9	286.0477	286.0481	−0.3	[M+H]^+^	285.04007, 117.03515, 159.04569, 143.05101, 93.03499
ESI+	Morin	11.01	302.0428	303.0499	0.4	[M+H]^+^	151.00266, 148.053, 149.02465
Flavones							
ESI–	Apigenin	8.4	270.0528	270.0528	−0.2	[M-H]^−^	149.0454, 269.0455, 151.004
ESI+	Luteolin	9.65	286.0474	286.0477	−1.2	[M+H]^+^	151.00371, 133.02939, 107.01382, 175.03984
ESI–	Hyperoside	9.21	464.0956	464.0951	0.2	[M-H]^−^	300.02724, 271.0244, 301.0343, 255.02955, 302.03781
Flavanones							
ESI–	Pinocembrin	11.05	256.0733	256.0733	−1	[M-H]^−^	255.02989, 227.03493, 151.00418
ESI–	Naringenin	11.77	272.0683	272.0684	−0.5	[M-H]^−^	271.06095, 119.05018, 151.00262, 177.01914
Other compounds and other polyphenols							
ESI–	Glucuronic acid	0.76	194.0426	194.0423	−1.7	[M-H]^−^	113.02403, 85.02986, 71.01426, 59.01395
ESI–	Phenylacetic acid	9.67	136.0524	136.0522	−0.8	[M-H]^−^	135.04485, 134.03748, 107.05043, 136.04749, 109.02961
ESI–	Resveratrol	9.12	228.0781	228.0784	−2.3	[M-H]^−^	225.1599, 227.07066, 183.02924
ESI–	Catechol	9.67	110.0365	110.0365	−2.6	[M-H]^−^	109.02961, 108.02031, 91.01858

Da: Dalton; ESI: electrospray ionization; ppm: parts per million; RT: retention time.

## Data Availability

Data will be available upon reasonable request.

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
