# Peer review of "Comparison of Phenolic Compounds and Evaluation of Antioxidant Properties of Porophyllum ruderale (Jacq.) Cass (Asteraceae) from Different Geographical Areas of Queretaro (Mexico)"

_plants, 2023, doi:10.3390/plants12203569_

Round 1

Reviewer 1 Report

Manusctript "Comparison of Phenolic Compounds and Evaluation of Antioxidant Properties of Porophyllum ruderale (Jacq.) Cass (Asterceae) from different geographical areas of Queretaro (Mexico)" is well written, interesting and would be valuable for the other scientists. 

However I have some comments and questions for the authors.

1. In my opinion, the data from section  2.1 section should be included in the section 4. Materials and methods.

2. Correct number of the table "UHPLC-DAD-QToF/MS-ESI characterization of identified phenolic compounds and other components in P. ruderale leaves from Landa de Matamoros and Arroyo Seco collection sites". It should be table 2, not table 1. Moreover, check the classification in this table. There are several errors, for example, myricetin, kaempferol and its glycosides are classified as a flavone, and hyperoside is classified as other compounds (please, check this classification in the other tables and in the manuscript too). In addition to this, remove notes under this table, as they are form Table 1.

3. What is the reason that (+)-catechin and (-)-epicatechin were not identified by UHPLC-DAD-QToF/MS-ESI if they are one of the main compounds in the phytochemical profile?

4. There are no eigenvalues in Supplementary Table S2.

5. Authors state that 2 principal components were used, but there are 3 principal component in Figure 3A, also these values do not match with values in Figure 3B.

6. Not all abbreviations of determined values are seen in Figure 3B

7. Line 168: Should be figure 3B

8. What could have caused dispersion of samples from Arroyo Seco along principal component 2?

9. There is no Figure 4, which is mentioned in the manuscipt.

10. Why hydroxycinnamic acids were not identified and quantified at wavelengths were are their spectrum maximum?

11. How many samples were collected in each location? 

Author Response

Reviewer 1

  1. Reviewer: Manuscript "Comparison of Phenolic Compounds and Evaluation of Antioxidant Properties of Porophyllum ruderale(Jacq.) Cass (Asteraceae) from different geographical areas of Queretaro (Mexico)" is well written, interesting and would be valuable for the other scientists. 

However I have some comments and questions for the authors.

In my opinion, the data from section  2.1 section should be included in the section 4. Materials and methods.

  • Authors’ response: Thank you for your comments, we hope our answers to all comments will solve all concerns from the reviewer. We have added the data from section 2.1 to section 4. Materials and methods. Please refer to the revised manuscript.

Revised Manuscript: Page 9. Lines 258-275.

  1. Reviewer: Correct number of the table "UHPLC-DAD-QToF/MS-ESI characterization of identified phenolic compounds and other components in  ruderaleleaves from Landa de Matamoros and Arroyo Seco collection sites". It should be table 2, not table 1. Moreover, check the classification in this table. There are several errors, for example, myricetin, kaempferol and its glycosides are classified as a flavone, and hyperoside is classified as other compounds (please, check this classification in the other tables and in the manuscript too). In addition to this, remove notes under this table, as they are form Table 1.
  • Authors’ response: Thank you. The labeling of all tables was rearranged. In Table 2, the description of the table was changed, and the classification of compounds was carried out according to “database on polyphenol content in foods”, from the website http:// phenol-explorer.eu/ in Table 2 and Supplementary Table S1. Please refer to the revised manuscript.

Revised manuscript:

  • Page 3, line 93: Table 1. Spectrophotometric total phenolic compounds content and antioxidant capacity of Porophyllum ruderale samples from three different geographical areas of Queretaro.”
  • Page 3, lines 105-107: Table 2. UHPLC-DAD-QToF/MS-ESI characterization of identified phenolic compounds and other components in ruderale leaves from Landa de Matamoros and Arroyo Seco collection sites.”
  • Page 2: “Supplementary Table S1. Content of phenolic compounds quantified by HPLC-DAD for P. ruderale from Landa de Matamoros and Jalpan de Serra.”
  1. Reviewer: What is the reason that (+)-catechin and (-)-epicatechin were not identified by UHPLC-DAD-QToF/MS-ESI if they are one of the main compounds in the phytochemical profile?
  • Authors’ response: Thanks for the observation. There are other studies where some polyphenolic compounds have also been overestimated with different matrix detectors. This is not clearly specified why it happens; however, it may be due to different factors, including a possible coelution phenomenon (Verdu et al., 2013). We are currently working on this phenomenon by refining the sample extraction and identification.

Verdu, C. F., Gatto, J., Freuze, I., Richomme, P., Laurens, F., & Guilet, D. (2013). Comparison of two methods, UHPLC-UV and UHPLC-MS/MS, for the quantification of polyphenols in cider apple juices. Molecules, 18(9), 10213-10227.

  1. Reviewer: There are no eigenvalues in Supplementary Table S2.
  • Authors’ response: The reviewer is correct. We have changed the name of this table. Please refer to the revised Supplementary Table.

Revised supplementary file:

Supplementary Table S2. Principal components, percentual participation, and cumulative per-cent. 

  1. Reviewer: Authors state that 2 principal components were used, but there are 3 principal component in Figure 3A, also these values do not match with values in Figure 3B.
  • Authors’ response: Thanks for the observations, we apologize with the reviewer as we did not state correctly what we wanted to express. In this section, we wanted to indicate that 2 of the components were enough to explain more than 80 % of the total variation (particularly: 97.78 %). Moreover, we did not use data from PC3, but we wanted to indicate is as additional information to the readers. We have clarified this in the revised manuscript.

Revised Manuscript:

Page 6, Lines 144-145: “(…) Figure 3, where two of the principal components (PC) explained 97.78% of the total variation in the samples”.

Page 6, Lines 156-158: “Although three components were indicated in Figure 3A, only two of them were plotted in Figure 3B as just two of them were enough to explain >80 % of the total variation.”.

  1. Reviewer: Not all abbreviations of determined values are seen in Figure 3B.
  • Authors’ response: Unfortunately, the software cannot provide a more enlarged figure, as indicated in the following figure:

However, we have tried to add the missing values in Fig. 3B. Please refer to the revised manuscript.

  1. Reviewer: Line 168: Should be figure 3B
  • Authors’ response: Dear reviewer, we appreciate your observation, the aforementioned correction was made. Page 6, line 150: reference is made to figure 3B.
  1. Reviewer: What could have caused dispersion of samples from Arroyo Seco along principal component 2?
  2. Authors’ response: Variation could be generated due to contrasting concentrations of TPC and some compounds quantified by HPLC (e.g., Epicatechin gallate), that are notoriously different from the other samples. Variations could be associated to the agroclimatic conditions of collection site, as Arroyo Seco has a semi-warm weather, whereas Landa de Matamoros is representative from a sub-humid environment. We have added this information in the revised manuscript.

Revised manuscript:

Page 6, Lines 158-159: Moreover, dispersion of the samples along PC2 (Figure 3B) could be associated to its contrasting TPC and selected HPLC-DAD quantified contents.

  1. Reviewer: There is no Figure 4, which is mentioned in the manuscript.
  • Authors’ response: Thank you for your observation, dear reviewer. The labeling of all figures was rearranged. Please refer to the revised manuscript.
  1. Reviewer: Why hydroxycinnamic acids were not identified and quantified at wavelengths were are their spectrum maximum?
  • Authors’ response: This may be due to the complexity and nature of the plant sample, as well as the extracting solvent that was used (ethanol 80%). However, the detection range of hydroxycinnamic acids is between wavelengths of 320 nm. In the present study, more hydroxycinnamic acids were detected compared to the study by Santiago-Saenz et al. (2018).

Reference:

Santiago-Saenz, Y. O., Hernández-Fuentes, A. D., Monroy-Torres, R., Cariño-Cortés, R., & Jiménez-Alvarado, R. (2018). Physicochemical, nutritional and antioxidant characterization of three vegetables (Amaranthus hybridus L., Chenopodium berlandieri L., Portulaca oleracea L.) as potential sources of phytochemicals and bioactive compounds. Journal of Food Measurement and Characterization, 12, 2855-2864.

  1. Reviewer: How many samples were collected in each location?
  • Authors’ response: In all cases, a total of 15 samples were collected in areas associated with seasonal agricultural production, in the dry season of the Sierra Gorda region, for which the samples collected were associated with a humid climate and a dry climate respectively. Please refer to the Revised Manuscript.

Revised Manuscript:

Page 10, lines 280-296: Porophyllum ruderale (Jacq.) Cass. (Asteraceae) [The International Compositae Alliance (TICA) checklist record: D78AD427-6E4D-4C8C-8C6A-BF2EDC6EA06C] fresh leaves were collected from three localities in the State of Querétaro (Mexico): Tlacote El Bajo (municipality of Querétaro ) [GPS coordinates: Longitude (dec): -100.507222, Latitude (dec): 20.662222] at 1850 m.a.s.l.; the town of Tlacote belongs to the municipality of Querétaro, the vegetation corresponds to xeric scrub with an annual precipitation of 224.40 mm in the year 2022, with a semi-dry and semi-warm climate [25]; Acatitlán de Zaragoza (municipality of Landa de Matamoros) [GPS coordinates: Longitude (dec): -99.188056, Latitude (dec): 21.206389] at 1200 meters above sea level (m.a.s.l.); with a cloud forest vegetation, having an average annual precipitation of 491.70 mm (2022), an altitude of 1288 m.a.s.l. and a semi-warm subhumid climate [26]. Lastly, El Aguacate (municipality of Arroyo Seco) [GPS coordinates: Longitude (dec): -99.616667, Latitude (dec): 21.399722] at 740 m.a.s.l.; its vegetation is low deciduous forest and xeric vegetation with an annual precipitation of 274 mm (2022) and a semi-dry climate [25,27]. For all cases, a total of 15 samples were collected in areas associated with seasonal agricultural production, in the dry season (spring/summer) of the Sierra Gorda and Querétaro region, for which the samples collected were associated with a humid climate and a dry climate respectively.

Reviewer 2 Report

The study presents the phenolic compounds and antioxidant capacity of Porophyllum ruderale plants from different locations in Queretaro, Mexico, revealed that plants from Queretaro exhibited lower levels of phenolic compounds, flavonoids, and antioxidant capacity when compared to plants from other sites. The predominant phenolic compounds identified were epicatechin and epigallocatechin gallate. The findings of this study suggest the importance of selecting plants with favorable agroclimatic conditions to achieve desired polyphenolic compositions and potential health benefits.

Considering the content and objectives of the study, it would be suitable for publication in a plants journal. However, certain modifications are necessary to improve the article. The recommended modifications include:

1.         Lines 108 and 119, there are two tables are written as Table 1 (Table 1. Spectrophotometric total phenolic compounds content and antioxidant capacity of Porophyllum ruderale samples from three different geographical areas of Queretaro.; Table 1. UHPLC-DAD-QToF/MS-ESI characterization of identified phenolic compounds and other components in P. ruderale leaves from Landa de Matamoros and Arroyo Seco collection sites.). Please revise.

2.         In the article, it is mentioned that the growing environment plays a crucial role as an influencing factor. To gain a more objective understanding of the environmental impact, it would be beneficial to include specific data regarding the growing environment of the samples. This could include information such as precipitation levels, solar irradiance, global temperature, and other relevant factors. Additionally, it is important to provide details about the planting date and harvesting date of the crops. It is necessary to determine whether the crops were harvested in the same season or not, as climate variations across different seasons can significantly affect plant growth.

3.         Lines 121-126, "Different letters by row express significant differences by Tukey-Kramer's test." no letters indicating significant differences were found in the table. letters indicating significant differences, please check it.

4.         In line 188, there is a reference to "Figure 4," but no corresponding figure with that label is manuscript. Please check it.

5.         Please add the LC-MS chromatograms of reference compound and sample solution?

6.         Please add the standard curves of reference compounds.

7.         Please add all vendors of reagents, such as the reagents used in mobile phase (lines 316-317).

8.         In line 95 and line 143 are all labeled as "Figure 1." Please revise it.

9.         The full names of abbreviations should appear at their first occurrence in paper. For example, the full names of HPLC and DAD are mentioned in lines 310-312, but the first occurrence of HPLC and DAD is in line 115.

Author Response

Reviewer 2:

  1. Reviewer: The study presents the phenolic compounds and antioxidant capacity of Porophyllum ruderale plants from different locations in Queretaro, Mexico, revealed that plants from Queretaro exhibited lower levels of phenolic compounds, flavonoids, and antioxidant capacity when compared to plants from other sites. The predominant phenolic compounds identified were epicatechin and epigallocatechin gallate. The findings of this study suggest the importance of selecting plants with favorable agroclimatic conditions to achieve desired polyphenolic compositions and potential health benefits.

Considering the content and objectives of the study, it would be suitable for publication in a Plants journal. However, certain modifications are necessary to improve the article. The recommended modifications include:

Lines 108 and 119, there are two tables are written as Table 1 (Table 1. Spectrophotometric total phenolic compounds content and antioxidant capacity of Porophyllum ruderale samples from three different geographical areas of Queretaro.; Table 1. UHPLC-DAD-QToF/MS-ESI characterization of identified phenolic compounds and other components in P. ruderale leaves from Landa de Matamoros and Arroyo Seco collection sites.). Please revise.

  • Authors’ response: Dear reviewer, thank you for your comments. There was indeed that error in the manuscript, which has already been modified. Please refer to the revised manuscript.
  1. Reviewer: In the article, it is mentioned that the growing environment plays a crucial role as an influencing factor. To gain a more objective understanding of the environmental impact, it would be beneficial to include specific data regarding the growing environment of the samples. This could include information such as precipitation levels, solar irradiance, global temperature, and other relevant factors. Additionally, it is important to provide details about the planting date and harvesting date of the crops. It is necessary to determine whether the crops were harvested in the same season or not, as climate variations across different seasons can significantly affect plant growth.
  • Authors’ response: Dear reviewer, thank you for your valuable recommendation. In relation to the planting of the plant sample, since Porophyllum ruderale is a plant that grows wild, it was not collected within any specific crop and a planting date has not been verified, however, some variables were considered. Agronomic information for harvest such as plant height, maturity, and the same harvest season. The plant samples were collected in areas associated with rainfed agricultural production, in the dry season of the Sierra Gorda region, for which the samples collected were associated with a humid climate and a dry climate respectively. Please refer to the revised manuscript.

Revised manuscript:

Page 10, Lines 280-296: Porophyllum ruderale (Jacq.) Cass. (Asteraceae) [The International Compositae Alliance (TICA) checklist record: D78AD427-6E4D-4C8C-8C6A-BF2EDC6EA06C] fresh leaves were collected from three localities in the State of Querétaro (Mexico): Tlacote El Bajo (municipality of Querétaro ) [GPS coordinates: Longitude (dec): -100.507222, Latitude (dec): 20.662222] at 1850 m.a.s.l.; the town of Tlacote belongs to the municipality of Querétaro, the vegetation corresponds to xeric scrub with an annual precipitation of 224.40 mm in the year 2022, with a semi-dry and semi-warm climate [25]; Acatitlán de Zaragoza (municipality of Landa de Matamoros) [GPS coordinates: Longitude (dec): -99.188056, Latitude (dec): 21.206389] at 1200 meters above sea level (m.a.s.l.); with a cloud forest vegetation, having an average annual precipitation of 491.70 mm (2022), an altitude of 1288 m.a.s.l. and a semi-warm subhumid climate [26]. Lastly, El Aguacate (municipality of Arroyo Seco) [GPS coordinates: Longitude (dec): -99.616667, Latitude (dec): 21.399722] at 740 m.a.s.l.; its vegetation is low deciduous forest and xeric vegetation with an annual precipitation of 274 mm (2022) and a semi-dry climate [25,27]. For all cases, a total of 15 samples were collected in areas associated with seasonal agricultural production, in the dry season (spring/summer) of the Sierra Gorda and Querétaro region, for which the samples collected were associated with a humid climate and a dry climate respectively.

  1. Reviewer: Lines 121-126, "Different letters by row express significant differences by Tukey-Kramer's test." no letters indicating significant differences were found in the table. letters indicating significant differences, please check it.
  • Authors’ response: Thank you, we have corrected this. Please refer to the revised manuscript.
  1. Reviewer: In line 188, there is a reference to "Figure 4," but no corresponding figure with that label is manuscript. Please check it.
  • Authors’ response: Thanks, modifications were made to the numbering of the figures. Please refer to the Revised manuscript.
  1. Reviewer: Please add the LC-MS chromatograms of reference compound and sample solution?
  • Authors’ response: Some significant chromatograms were added in the supplementary part page 3. Please refer to the revised Supplementary File.
  1. Reviewer: Please add the standard curves of reference compounds.
  • Authors’ response: Standard curves were added to the Supplementary Information. Please refer to the Revised Supplementary File in Supplementary Table S3. This table was also indicated in Page 11, Line 349.
  1. Reviewer: Please add all vendors of reagents, such as the reagents used in mobile phase (lines 316-317).
  • Authors’ response: The standards used for HPLC analysis were HPLC grade reagents ≥95%, solids which were purchased from Sigma-Aldrich. On the other hand, formic acid and acetonitrile were purchased by the company “Prolab”.

Revised Manuscript:

Page 10, line 327-328: “(…) both HPLC-grade reagents (Prolab company, Qro Mexico).

Revised manuscript:

  1. Reviewer: In line 95 and line 143 are all labeled as "Figure 1." Please revise it.
  • Authors’ response: Dear reviewer, we appreciate your valuable observation, indeed the labeling of the figures was not correctly in order, the arrangement has been made. Please refer to the revised manuscript.
  1. Reviewer: The full names of abbreviations should appear at their first occurrence in paper. For example, the full names of HPLC and DAD are mentioned in lines 310-312, but the first occurrence of HPLC and DAD is in line 115.
  • Authors’ response: Dear reviewer, according to your observation, the names of the abbreviations of the colorimetric techniques of antioxidant capacity DPPH, FRAP and ABTS, as well as HPLC and UHPLC-DAD-QToF/MS-ESI, found in section 2.2, were added.

Revised manuscript:

Page 2, line: 89-9: “(…) the highest 2,2-diphenyl-1-picrylhydrazil (DPPH), ferric reducing antioxidant power (FRAP), and 2,2- azino-bis(3-ethylbenzothiazoline-6-sulfonic acid) (ABTS) antioxidant capacity.

Page 3, line: 100-102: “Table 2 shows the ultra-high performance chromatographic analysis, coupled with diode-array-detector, quadrupole time of-flight, and mass spectrometry with electrospray ionization(UHPLC-DAD-QToF/MS-ESI)”

Page 4, line 111: high-performance liquid chromatography (HPLC).”
